# In Situ PD-L1 Expression in Oral Squamous Cell Carcinoma Is Induced by Heterogeneous Mechanisms among Patients

**DOI:** 10.3390/ijms23084077

**Published:** 2022-04-07

**Authors:** Yutaro Kondo, Susumu Suzuki, Shoya Ono, Mitsuo Goto, Satoru Miyabe, Tetsuya Ogawa, Hiromi Tsuchida, Hideaki Ito, Taishi Takahara, Akira Satou, Toyonori Tsuzuki, Kazuhiro Yoshikawa, Ryuzo Ueda, Toru Nagao

**Affiliations:** 1Department of Maxillofacial Surgery, Aichi Gakuin University School of Dentistry, Nagoya 464-0821, Japan; kndytr@gmail.com (Y.K.); sono891002@gmail.com (S.O.); mgoto@dpc.agu.ac.jp (M.G.); smiyabe@dpc.agu.ac.jp (S.M.); tnagao@dpc.agu.ac.jp (T.N.); 2Research Creation Support Center, Aichi Medical University School of Medicine, Nagakute 480-1195, Japan; htsuchid@gifu-u.ac.jp (H.T.); yoshikaw@aichi-med-u.ac.jp (K.Y.); 3Department of Tumor Immunology, Aichi Medical University School of Medicine, Nagakute 480-1195, Japan; uedaryu@aichi-med-u.ac.jp; 4Department of Otorhinolaryngology, Aichi Medical University School of Medicine, Nagakute 480-1195, Japan; ogawate@aichi-med-u.ac.jp; 5Department of Pathology, Aichi Medical University School of Medicine, Nagakute 480-1195, Japan; itou.hideaki.820@mail.aichi-med-u.ac.jp; 6Department of Surgical Pathology, Aichi Medical University Hospital, Nagakute 480-1195, Japan; takahara.taishi.456@mail.aichi-med-u.ac.jp (T.T.); satou.akira.442@mail.aichi-med-u.ac.jp (A.S.); tsuzuki@aichi-med-u.ac.jp (T.T.)

**Keywords:** OSCC, oral squamous cell carcinoma, EGF, epidermal growth factor, IFN-γ, interferon gamma, PD-L1, programmed death ligand-1

## Abstract

The expression of programmed death ligand-1 (PD-L1) is controlled by complex mechanisms. The elucidation of the molecular mechanisms of PD-L1 expression is important for the exploration of new insights into PD-1 blockade therapy. Detailed mechanisms of the in situ expression of PD-L1 in tissues of oral squamous cell carcinomas (OSCCs) have not yet been clarified. We examined the mechanisms of PD-L1 expression focusing on the phosphorylation of downstream molecules of epidermal growth factor (EGF) and interferon gamma (IFN-γ) signaling in vitro and in vivo by immunoblotting and multi-fluorescence immunohistochemistry (MF-IHC), respectively. The in vitro experiments demonstrated that PD-L1 expression in OSCC cell lines is upregulated by EGF via the EGF receptor (EGFR)/PI3K/AKT pathway, the EGFR/STAT1 pathway, and the EGFR/MEK/ERK pathway, and by IFN-γ via the JAK2/STAT1 pathway. MF-IHC demonstrated that STAT1 and EGFR phosphorylation was frequently shown in PD-L1-positive cases and STAT1 phosphorylation was correlated with lymphocyte infiltration and EGFR phosphorylation. Moreover, the phosphorylation pattern of the related molecules in PD-L1-positive cells differed among the cases investigated. These findings indicate that PD-L1 expression mechanisms differ depending on the tissue environment and suggest that the examination of the tissue environment and molecular alterations of cancer cells affecting PD-L1 expression make it necessary for each patient to choose the appropriate combination drugs for PD-1 blockade cancer treatment.

## 1. Introduction

Many types of cells, such as lymphocytes, macrophages, myeloid-derived suppressive cells, and fibroblasts, infiltrate into cancer tissues and release many kinds of cytokines and create a complex microenvironment called the tumor microenvironment (TME), which influences tumor development and progression and interferes with the efficacy of immunotherapy [1,2]. Programmed death ligand-1 (PD-L1) expressed in the TME is known to be an important molecule for effector T-cell suppression and PD-L1 in cancer cells, and stromal cells interact with programmed death-1 (PD-1) in effector T-cells, which results in the suppression of cancer immunity [3,4]. We previously reported that PD-L1 was expressed in head and neck cancer (HNC) with a high frequency [5]. Therefore, the elucidation of the PD-L1 expression mechanisms is important in developing new PD-1 blockade HNC immunotherapy.

Oral squamous cell carcinoma (OSCC) is the most common type of HNC, and anti-PD-1 inhibitors, such as nivolumab and pembrolizumab, have shown efficacy for HNC immunotherapy [6,7]. However, limited cases have shown this efficacy, and more investigation is needed regarding the mechanism of the expression of PD-L1, a ligand for PD-1.

The expression mechanisms of PD-L1 have been reported mainly as oncogenic stimuli, inflammatory cytokines, and mutated cancer-driver genes [8,9,10,11]. The expression of PD-L1 in tumors with abundant immune cell infiltration is thought to be due to exogenous stimulation through the Janus kinase–signal transducer activator of the transcription (JAK/STAT) pathway by the stimulation of interferon gamma (IFN-γ) [8,10,12,13]. In these types of tumors, the reactivation of exhausted lymphocytes by anti-PD-L1/PD-1 therapy is expected to be effective [8,14,15]. However, PD-L1 expression is also found in tumors with a poor infiltration of immune cells [14,15,16]. In these types of tumors, effects of endogenous oncogenic stimuli or genetic mutations can occur. The expression of PD-L1 in HNCs is thought to be related to endogenous stimulation by epidermal growth factor (EGF) and exogenous stimulation by IFN-γ [10]. Since EGF receptors (EGFRs) are overexpressed in most OSCCs [17], EGFR activation may be an important factor in regulating the expression of PD-L1 as well as IFN-γ. The downstream of EGFR is mainly composed of the STAT, Ras/Raf/mitogen-activated protein kinase (MAPK)/extracellular signal-regulated kinase (ERK), and phosphatidylinositol 3-kinase (PI3K)/Akt pathways [18]; however, there is still no unified view as to which pathway is the key regulator of PD-L1 expression [9,10,19,20,21]. Furthermore, there are few reports from immunohistochemical studies regarding the heterogeneity of PD-L1 expression mechanisms in OSCC tissues.

In this study, we examined the mechanisms of PD-L1 expression using OSCC cell lines focused on EGF and IFN-γ signaling. Furthermore, in PD-L1-positive OSCC tissues, the pattern of phosphorylation of PD-L1 expression-related molecules differed among the different cases. These findings indicate that the PD-L1 expression mechanisms are heterogeneous depending on the tissue environment and might provide new insights for OSCC immunotherapy.

## 2. Results

### 2.1. Upregulation of PD-L1 Expression in OSCC Cell Lines by EGF through the EGFR/PI3K/AKT, EGFR/MEK/ERK, and/or EGFR/STAT1 Pathways

The PD-L1 expression level in HSC3, HSC4, and SAS increased by EGF in a dose-dependent manner (Figure 1A). EGFR and its downstream molecules, STAT1, AKT, and ERK, were phosphorylated for 10 to 120 min after stimulation with EGF (Figure 1B). These inductions and phosphorylation were inhibited by the EGFR inhibitors, cetuximab, and gefitinib, which suggests that EGF upregulates PD-L1 expression in OSCC cell lines (Figure 1C,D). The upregulation of PD-L1 by EGF was inhibited by PI3Ki (Omipalisib) and MEKi (GSK1120212), accompanied by the inhibition of phosphorylation of AKT and ERK, respectively, in HSC3 (Figure 1E). Interestingly, PD-L1 upregulation was inhibited only by PI3Ki, but not by MEKi in HSC4, and, contrarily, it was inhibited only by MEKi but not PI3Ki in SAS. JAK2i (AZD1480) suppressed PD-L1 expression at high concentrations (10 μM) only in HSC3. However, it was not suppressed at low concentrations, and phosphorylation of STAT1 was not inhibited, suggesting the occurrence of non-specific inhibition. These findings suggest that PD-L1 expression was upregulated by EGF through the EGFR/PI3K/AKT, EGFR/MEK/ERK, and/or EGFR/STAT1 pathways. However, the pathways utilized were dependent on the cells (Figure 2D).

### 2.2. Upregulation of PD-L1 Expression in OSCC Cell Lines by IFN-γ through the JAK2/STAT1 Pathway

The PD-L1 expression levels in HSC3, HSC4, and SAS increased with IFN-γ in a dose-dependent manner (Figure 2A), and the phosphorylation of IFN-γ downstream molecules, STAT1, was induced at 10 to 120 min after the stimulation with IFN-γ; however, the EGFR downstream molecules AKT and ERK were not phosphorylated (Figure 2B). PD-L1 induction and STAT1 phosphorylation were inhibited by JAK2i (AZD1480) (Figure 2C), which suggests that IFN-γ upregulates PD-L1 expression in OSCC cell lines through the JAK2/STAT1 pathway (Figure 2D).

### 2.3. MF-IHC Staining of HSC3 Cells and OSCC Tissues

We detected PD-L1 expression and phosphorylation of the downstream molecules, i.e., EGFR, AKT, and STAT1, at 24 h after stimulation with EGF or IFN-γ by MF-IHC using HSC3 paraffin-embedded sections (Figure 3). The phosphorylation pattern was compared with the results of the Western blot. PD-L1 upregulation was observed after the stimulation of EGF or IFN-γ. Only STAT1 phosphorylation was observed in the stimulation with IFN-γ. In contrast, phosphorylation of all the investigated molecules, i.e., EGFR, AKT, and STAT1, was observed in the stimulation of EGF. These results on phosphorylation patterns were consistent with those of the Western blot, indicating that the analysis of the phosphorylation pattern by MF-IHC can deduce the in situ pathway. Furthermore, the EGF pathway or the IFN-γ pathway was used for PD-L1 expression.

Representative MF-IHC staining images of clinical tissues are shown in Figure 4. Phosphorylation of AKT was undetectable in all of samples (data not shown), despite AKT phosphorylation being unable to be detected in the model case using HSC3 (Figure 3).

Anti-PD-L1 and anti-p-EGFR stained the plasma membrane, and anti-p-STAT1 stained the nucleus (Figure 4A–C). Lymphocyte infiltration, as a source of IFN-γ, was determined by tumor immune microenvironment (TIME) classification: “Desert” (Figure 4D), “Excluded” (Figure 4E), and “Inflamed” (Figure 4F), according to previous reports [14,15].

### 2.4. Relationship between PD-L1 Expression, Phosphorylation of EGFR and STAT1, and Lymphocyte Infiltration in OSCC Tissues

We examined the relationship between PD-L1 expression, phosphorylation of EGFR and STAT1, and lymphocyte infiltration in OSCC tissues. PD-L1 expression was observed in 23 of 50 patients (46%). The p-STAT1 score increased significantly in the PD-L1-positive group (Figure 5A), and the p-EGFR score showed an increasing trend, although this was not significant (Figure 5B). Moreover, the p-STAT1 score was increased in the “Inflamed” group compared to the “Desert” group (Figure 5C). Increased PD-L1 expression was observed in the “Inflamed” group compared to that in the “Desert” group, suggesting that PD-L1 expression by IFN-γ derived from infiltrated lymphocytes (Figure 5D).

In Figure 5E, the relationship between the p-EGFR and p-STAT1 scores, PD-L1 expression, and lymphocyte infiltration is shown. The p-EGFR and p-STAT1 scores showed a positive correlation, suggesting EGFR activation-induced phosphorylation of STAT1 in the OSCC tissues. When the samples were divided to four regions, the following was observed: a p-STAT1 score of 0–1 and a p-EGFR score of 0–1 (I); a p-STAT1 score of 0–1 and a p-EGFR score of 2–4 (II); a p-STAT1 score of 2–4 and a p-EGFR score of 0–1 (III); a p-STAT1 score of 2–4 and a p-EGFR score of 2–4 (IV); and PD-L1-positive (>1%) cases were observed in 20.0% of I (3/15), 0% of II (0/0), 59.1% of III (13/22), and 53.8% of IV (7/13). Most of the PD-L1-positive cases were observed in regions III and IV. The cases in region III and IV were further divided to “Inflamed” cases and “Excluded+Desert” cases. The “Inflamed” cases in region IV indicate that PD-L1 expression is affected by both EGF and IFN-γ. The “Excluded+Desert” cases in region IV indicate that PD-L1 expression is affected by EGF more than IFN-γ, and the “Inflamed” cases in region III indicate that PD-L1 expression is affected by IFN-γ more than EGF. PD-L1-positive cases were observed in all the “Inflamed” cases in region IV (5/5), in 25% of the “Excluded+Desert” cases in region IV (2/8), and in 54% of the “Inflamed” cases in region III (6/11). These results suggest that IFN-γ is more effective than EGF for PD-L1 expression, and EGF might enhance IFN-γ-induced PD-L1 expression. Referring to the TCGA HNSC tumor dataset, we found a positive correlation between IFN-γ-related genes (IFN-γ, CD8A, and STAT1) and PD-L1 (CD274), but not between EGF-related genes (EGFR, AKT1, and MAP2K7), which seems to support the results of this study (Figure 6).

## 3. Discussion

It is known that PD-L1 expression in cancer cells is controlled by both oncogenic pathways (OncoPath) and immunologic pathways (ImmunoPath) in OSCCs [10]. We confirmed previous findings using OSCC cell lines, i.e., HSC3, HSC4, and SAS in vitro (Figure 1 and Figure 2). EGF is a representative factor driving the OncoPath, and IFN-γ is a representative factor driving ImmunoPath-induced PD-L1 expression via the signaling cascade of EGFR/PI3K/AKT, EGFR/STAT1, or EGFR/MEK/ERK for EGF and of JAK2/STAT1 for IFN-γ, respectively. However, the EGFR downstream cascade differed among the cell lines. The phosphorylation pattern of the downstream cascade molecules in the HSC3 cells stimulated with EGF or IFN-γ using MF-IHC analysis (Figure 3) was consistent with the results of the Western blot, suggesting that phosphorylation analysis in situ using MF-IHC is feasible.

Our findings (Figure 5) suggest that the ImmunoPath is important for PD-L1 expression in OSCC tissues, more so than the EGF-induced OncoPath (EGF OncoPath). More than 90% of HNCs overexpress EGFR [22]; this may be related to PD-L1 expression. However, there was a difference between EGFR expression and its activation in HNCs [23,24], as the actual number of cases in which EGFR activation increases PD-L1 may be less than expected. Additionally, our findings also suggest that the PD-L1 expression mechanisms in OSCC tissues are heterogenous and can be classified into five types: both the ImmunoPath and EGF OncoPath are inactive (Type 1, Region I in Figure 5E); the ImmunoPath is dominantly active (Type 2, “Inflamed” cases of Region III in Figure 5E); the EGF OncoPath is dominantly active (Type 3, “Excluded+Desert” cases of Region IV in Figure 5E); both the ImmunoPath and EGF OncoPath are active (Type 4, “Inflamed” cases of Region IV in Figure 5E); and the remaining types (Type 5, “Excluded+Desert” cases of Region III in Figure 5E). The OncoPath excluding EGF is thought to affect PD-L1 expression in Type 1 and Type 5. For example, c-MET activation, PTEN loss, and PI3KCA mutations are candidates [25,26]. Elucidating the mechanism of PD-L1 expression in these types will be a future subject of investigation.

It is known that TME affects the response to immune checkpoint therapies, including PD-1 blockade [14,15,27]. Our classification above may contribute to predictions for PD-1 blockade therapies and the development of combination therapies using the PD-1 blockade and novel immune-checkpoint therapies. Our findings indicate that Type 2 and 4 are effective, but therapeutic effects are not expected in Type 1, 3, and 5. It is reported that EGFR-mutated NSCLC forms an immunosuppressive TME by suppressing effector T-cell infiltration and inducing Treg via the EGFR downstream molecules AKT1 and JNK [28]. The regulation of the EGF OncoPath is thought to be necessary in Type 3 for effector T-cell infiltration [29], and also in Type 4 because the EGF OncoPath might involve the enhancement of ImmunoPath-induced PD-L1 expression. For example, pembrolizumab used in combination with cetuximab has shown great efficacy in a Phase Ⅱ trial of recurrent or metastatic HNCs and is considered a good candidate [30]. Neither the EGF OncoPath nor ImmunoPath are active in Type 1 and 5; therefore, other therapeutic strategies might be necessary.

There are several limitations to this study. First, this was a retrospective study with a small number of cases. In addition, since we used tissues that were as fresh as possible in consideration of antibody staining, we were not be able to examine the relationship with prognosis. A prospective study with a larger number of cases is necessary in the future. Furthermore, many mechanisms for the regulation of PD-L1 expression are already known, but we investigated the regulation of PD-L1 only at the transcriptional level focused on the EGF and IFN-γ downstream cascade, and we only investigated the phosphorylation of EGFR and STAT1 in situ. Finally, in this study, we only investigated PD-L1 expression in tumor cells. PD-L1 expression in OSCCs is found in stromal immune cells, such as macrophages and tumor cells [5]. Furthermore, in contrast to tumor cells, PD-L1 expression in macrophages has been reported to have a favorable prognosis [31,32]. Further comprehensive research is needed to better understand the expression mechanism and function of PD-L1 in both cancer cells and immune cells.

In conclusion, understanding OncoPath and ImmunoPath which regulate PD-L1 expression in OSCCs is critical for prognostic biomarkers and patient selection for PD-1 blockade therapy.

## 4. Materials and Methods

### 4.1. OSCC Cell Lines

Human OSCC cell lines (HSC3, HSC4, and SAS) were preserved at the Research Creation Support Center of Aichi Medical University, in Nagakute, Japan. Cells were maintained in DMEM (Gibco, Paisley, UK) and supplemented with 10% fetal bovine serum (FBS; HyClone Laboratories, Inc., South Logan, UT, USA) and 1% penicillin–streptomycin (Gibco, Grand Island, NY, USA) in 75 mL flasks at 37 °C and 5% CO_2_ humidified air.

### 4.2. Clinical Samples

Fifty OSCC patients who underwent surgical resection at the Department of Maxillofacial Surgery, Aichi Gakuin University (Nagoya, Japan) and who had not received neoadjuvant chemotherapy or radiation therapy were included in the study. Demineralized samples were analyzed using preoperative biopsy tissue. This study was conducted in accordance with the Declaration of Helsinki and approved by the Ethical Committee of Aichi Gakuin University (approval number: 82, approval date: 24 June 2019) and Aichi Medical University (approval number: 2020-H033, approval date: 25 May 2020).

### 4.3. Cell Treatment and Reagents

OSCC cell lines were cultured in 6-well plates (2 × 10^5^ cells/well) for 2 days and starved for 5 days to exclude the potential effect of cytokines in the serum. To measure PD-L1 expression, cells were treated with EGF and IFN-γ at the indicated dose for 24 h in the presence of following inhibitors: cetuximab (Merck, Kenilworth, NJ, USA), gefitinib (Tocris Bioscience, Avonmouth, Bristol, UK), AZD1480 (JAK2i, Selleckchem, Houston, TX, USA), Omipalisib (PI3Ki, Selleckchem), and GSK1120212 (MEKi, Selleckchem). Then, PD-L1 expression in OSCC cell lines was measured by flow cytometry.

### 4.4. Flow Cytometry

Cells were harvested with Accutase (Innovative Cell Technologies, Inc., San Diego, CA, USA) after cell treatment and stained for 20 min at 4 °C with mouse anti-human CD274 (PD-L1) mAb, followed by incubation with FITC-conjugated anti-mouse IgG (MBL, Tokyo, Japan) for 20 min at 4 °C. Normal mouse IgG was used as a negative control. Then, the cells were analyzed on a FACS CantoⅡ (BD Biosciences, San Jose, CA, USA) with the aid of Flow Jo software (Tree Star, Inc., Ashland, OR, USA). PD-L1 expression was calculated as follows: [anti-human PD-L1 mAb mean fluorescence intensity (MFI)—normal mouse IgG MFI].

### 4.5. Western Blotting

Aliquots of 1 × 10^6^ cells were lysed in 50 µL of ice-cold protein extraction buffer (2% Triton X in 10 mM Tris-HCl, 150 mM NaCl, 2 mM ethylenediaminetetraacetic acid (EDTA), and 2 mM 2-mercaptoethanol (pH 7.4)) for 30 min. The cell lysates were resolved on SDS-PAGE gels (10^5^ cells/lane) and transferred to nitrocellulose membranes. After blocking with tris-buffered saline (TBS) with 1% skim milk for 1 h at room temperature, the membranes were incubated with primary antibodies. The following primary antibodies were used: EGFR (1:1000 dilution, MBL), p-EGFR (Y1068, 1:1000 dilution, Cell Signaling Technology, Danvers, MA, USA), p-AKT (S473, 1:2000 dilution, Cell Signaling Technology), p-STAT1 (Y701, 1:1000 dilution, Cell Signaling Technology), p-ERK(1/2) (Thr202/Try204, 1:1000 dilution, Cell Signaling Technology), and β-actin (1:1000 dilution, MBL). β-actin was used as an internal control. Then, they were washed with TBS four times, incubated with peroxidase polymer anti-mouse IgG or anti-rabbit IgG (Vector Laboratories, Burlingame, CA, USA) at a 1:100 dilution for 30 min at room temperature. After washing with TBS four times, the protein signal was detected using ECL Prime Western blotting Detection Reagent (GE Healthcare Systems, Chicago, IL, USA) and imaged using an Amersham Imager 600 (GE Healthcare Systems).

### 4.6. Multi-Color Immunofluorescence Histochemistry (MF-IHC)

MF-IHC was performed on formalin-fixed paraffin-embedded OSCC tissue sections using the same procedure as that in our previous report [33]. The following primary antibodies were used: CD3 (clone M4622, Spring Biosciences, Pleasanton, CA, USA), CD8 (clone 1A5, Biogenex, Fremont, CA, USA), cytokeratin (AE1+AE3, Biogenex, Fremont, CA, USA), PD-L1 (Abcam, Cambridge, UK), p-STAT1 (Ser727, Cell Signaling Technology), and p-EGFR (Y1068+Y1092, Cell Signaling Technology). The multi-color staining images were captured using an Aperio CS2 (Leica Microsystems, Wetzlar, Germany) for the HSC3 cells or the Vectra system (PerkinElmer, Waltham, MA, USA) for the OSCC tissues.

### 4.7. Image Analysis

The slides were scanned with Vectra at a low resolution, and a region of interest (ROI) was set at the invasive front of the tumor where lymphocytes were accumulated with Phenochart software (PerkinElmer). Then, at least three ROIs were captured with a 20× objective, and the reconstructed multispectral images were obtained and the number of marker-positive cells was counted using Inform software (PerkinElmer). For PD-L1, p-STAT1- and p-EGFR-positive tumor cells were counted as double-positive cells with cytokeratin. To analyze the IHC results, a semi-quantitative method was employed based on previous reports [34,35]. Specifically, the IHC scores for p-EGFR and p-STAT1 were evaluated on a scale of 0–4 (0, <5%; 1, 5–25%; 2, 25–50%; 3, 50–75%; 4, >75%). For PD-L1, a positivity rate of 1% or higher was considered positive.

### 4.8. Database Analysis

Gene expression in OSCC was analyzed using the Gene Expression Profiling Interactive Analysis (GEPIA) online database (http://gepia.cancer-pku.cn accessed on 10 December 2021) [36]. The Cancer Genome Atlas (TCGA) HNSC tumor data were used as the basis for the correlation analysis.

### 4.9. Statistical Analysis

The numerical data were presented as the mean (±SE: standard error) of independent experiments, and differences were examined using the Tukey’s test. The Mann–Whitney U test and the Steel–Dwass test were performed to determine the significance of the IHC analysis. The correlation analysis data were determined using the Spearman’s rank correlation coefficient. Statistical analyses were performed using the R Statistical Software (version 3.6.3; Foundation for Statistical Computing, Vienna, Austria). *p* < 0.05 was considered statistically significant.

## Figures and Tables

**Figure 1 ijms-23-04077-f001:**
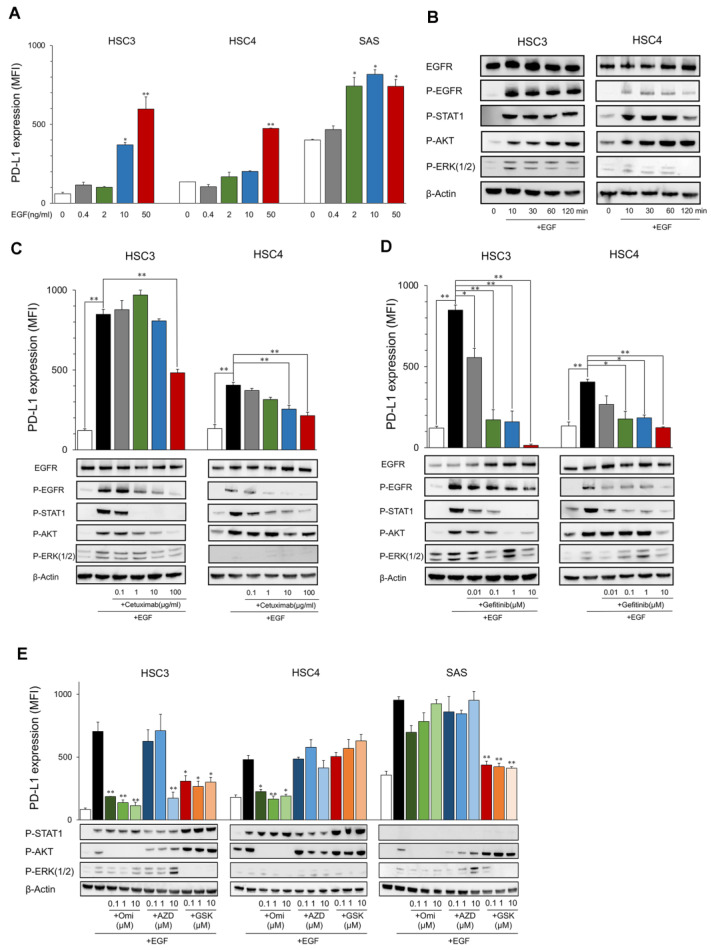
Expression of PD-L1 and phosphorylation of EGFR downstream proteins by EGF. (**A**) OSCC cell lines were treated with EGF at the indicated dose. PD-L1 expression was tested by flow cytometry. Statistical analysis was performed on the results of EGF 0 ng/mL. (**B**) Cells were treated with EGF (50 ng/mL) each time, and the phosphorylation of EGFR downstream proteins was evaluated by Western blotting. (**C**,**D**) Cetuximab and gefitinib were added into the OSCC cell culture at the indicated dose 60 min before EGF treatment (50 ng/mL). PD-L1 expression and the phosphorylation of EGFR downstream proteins were evaluated after 24 h and 60 min, respectively. (**E**) Omipalisib, AZD1480, and GSK1120212 were added into the OSCC cell culture at the indicated dose 60 min before EGF treatment (50 ng/mL). PD-L1 expression and phosphorylation of EGFR downstream proteins were evaluated after 24 h and 60 min, respectively. Statistical analysis was performed for the addition of EGF alone. * *p* < 0.05, ** *p* < 0.01.

**Figure 2 ijms-23-04077-f002:**
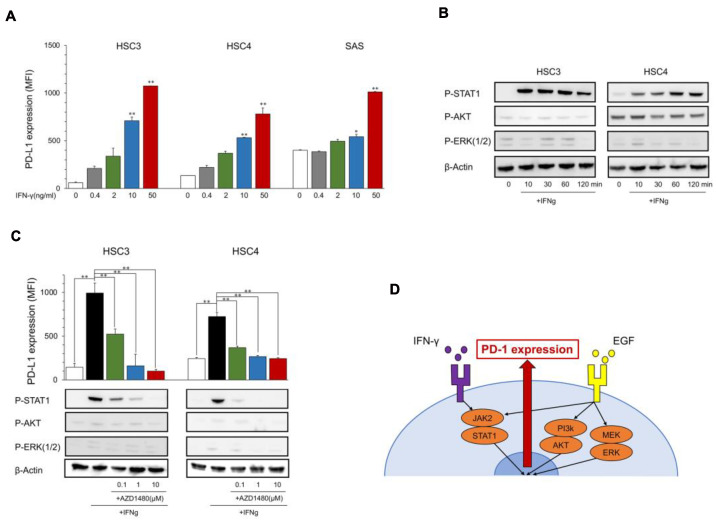
Expression of PD-L1 and phosphorylation of downstream proteins by IFN-γ. (**A**) OSCC cell lines were treated with IFN-γ at the indicated dose. PD-L1 expression was tested using flow cytometry. Statistical analysis was performed on the results of IFN-γ 0 ng/mL. (**B**) Cells were treated with IFN-γ (25 ng/mL) each time, and the phosphorylation of downstream proteins was evaluated by Western blotting. (**C**) AZD1480 was added into the OSCC cell culture at the indicated dose 60 min before IFN-γ treatment (25 ng/mL). PD-L1 expression and the phosphorylation of EGFR downstream proteins were evaluated after 24 h and 60 min, respectively. (**D**) A graphic summary of in vitro assays. * *p* < 0.05, ** *p* < 0.01.

**Figure 3 ijms-23-04077-f003:**
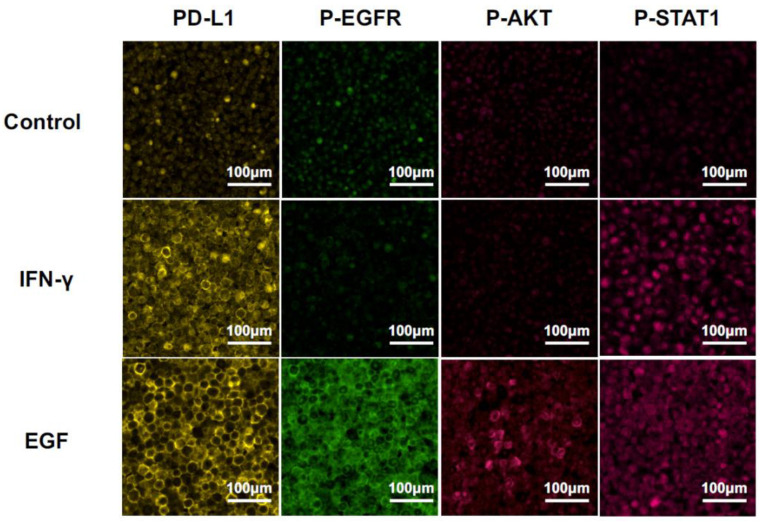
Changes in MF-IHC staining for PD-L1, p-EGFR, p-AKT, and p-STAT1 by EGF and IFN-γ stimulation of HSC3. HSC3 was stimulated with EGF or IFN-γ for 24 h, and we investigated changes in the MF-IHC staining for PD-L1, p-EGFR, p-AKT, and p-STAT1. Scale bars are indicated in each image.

**Figure 4 ijms-23-04077-f004:**
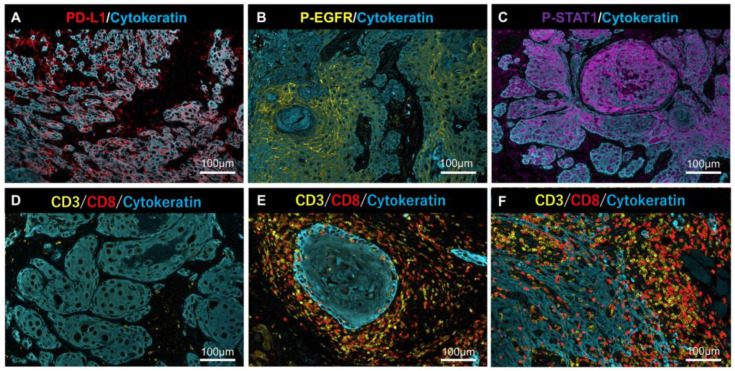
Representative MF-IHC staining results for PD-L1, p-EGFR, p-STAT1, and lymphocytes. Positive cases of PD-L1 (**A**), p-EGFR, (**B**) and p-STAT1 (**C**) in tumor cells. Representative case of (**D**) “Desert” (very few T-cells infiltrated into cancer tissues), (**E**) “Excluded” (T-cells infiltrated into stromal areas but not into the nest), and (**F**) “Inflamed” (T-cells infiltrated into the nest and stroma). Scale bars are indicated in each image.

**Figure 5 ijms-23-04077-f005:**
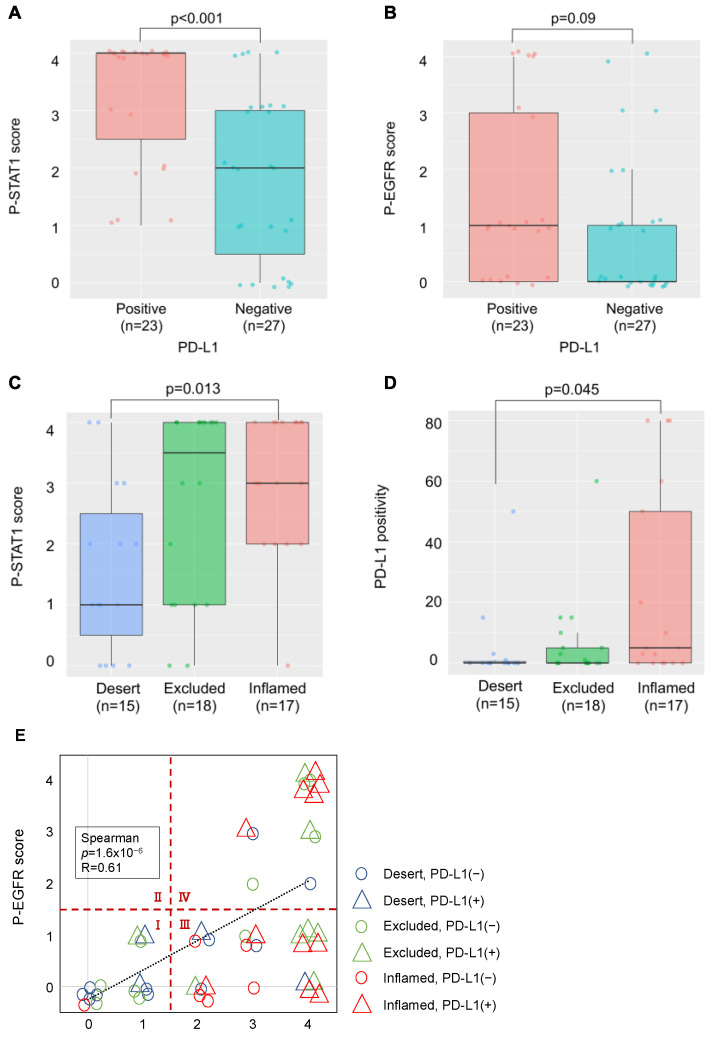
Relationship between PD-L1 expression, phosphorylation of PD-L1-related proteins, and lymphocyte infiltration in OSCC tissues. (**A**) Comparison of the p-STAT1 scores in PD-L1-positive and -negative cases. (**B**) Comparison of the p-EGFR scores in PD-L1-positive and -negative cases. (**C**) Comparison of the p-STAT1 scores for each TIME classification. (**D**) Comparison of PD-L1 positivity for each TIME classification. (**E**) Relationship between p-EGFR and p-STAT1 scores, PD-L1 expression, and TIME classification.

**Figure 6 ijms-23-04077-f006:**
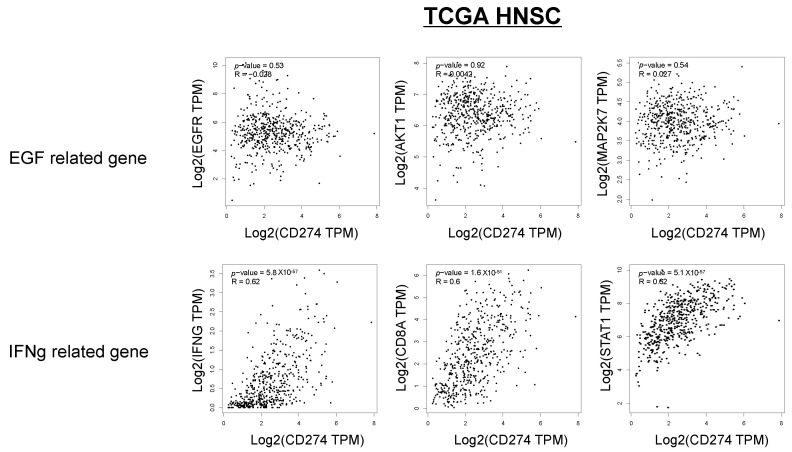
Correlation between IFN-γ-related genes, EGF-related genes, and PD-L1 expression in the TCGA HNSC dataset. Referring to the TCGA HNSC tumor dataset, we investigated the correlation between IFN-γ-related genes (IFN-γ, CD8A, and STAT1), EGF-related genes (EGFR, AKT1, and MAP2K7), and PD-L1 (CD274).

## Data Availability

The data that support the findings of this study are available from the corresponding author, S.S., upon reasonable request.

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
