# Peer review of "In Situ PD-L1 Expression in Oral Squamous Cell Carcinoma Is Induced by Heterogeneous Mechanisms among Patients"

_ijms, 2022, doi:10.3390/ijms23084077_

Round 1

Reviewer 1 Report

Dear Authors

Your paper reads well. It follows the work done in 2016 by R.L.Ferris team.

Only one question regarding the cut off of PDL1 staining and also the distribution ie. cytoplasmic or membranous.

Please try rephrase your introduction; be more clear about constituents of TIME.

Author Response

Only one question regarding the cut off of PDL1 staining and also the distribution ie. cytoplasmic or membranous.

Response; Thank you for your questions. For PD-L1, a positivity rate of 1% or higher was considered positive. Please refer last sentence of “4.7. Image Analysis” in page 10 (highlighted). And PD-L1 expression was shown on the plasma membrane. Please refer line 4 in page 5 (highlighted).

Please try rephrase your introduction; be more clear about constituents of TIME.

Response; Thank you for your comment. Although a term “Time Immune Microenvironment (TIME)” was used in Nat. Med. 2018, 24, 541–550 (ref no. 15) for the understanding of the complexity and diversity of the immune context of the tumor microenvironment, TIME is not appropriate term in the context of the introduction. Therefore, we changed “TIME” to Tumor Microenvironment (TME) to avoid misunderstanding for readers. Please refer the 4th line from the bottom of the 1st page to the 1st line of the 2nd page.

Reviewer 2 Report

The paper tried to suggest that the PD-L1 expression mechanisms were heterogeneous depending on the tissue environment and might provide new insights for OSCC immunotherapy.

The conceptual advancement gained by the present study was limited.

Major issues:

1.In Figure1 and Figure 2, the total protein levels of ERK, Akt and STAT1 need to be detected.

2.In the cell experiment, it is best to increase the WB detection of PD-L1.

Author Response

  1. In Figure1 and Figure 2, the total protein levels of ERK, Akt and STAT1 need to be detected.

Response; Thank you very much for your comment. We agree with your comment. Phosphorylation level of these proteins must be evaluated exactly by comparing total protein levels of each protein as a control. However, we investigated the phosphorylation levels comparing with β-actin as a house keeping molecule in this study. In the future, we would like to follow your comment and examine the phosphorylation level comparing with each protein as a control.

  1. In the cell experiment, it is best to increase the WB detection of PD-L1.

Response; We appreciate your comment. We evaluated the reactivity of 3 clones of PD-L1 antibody, SP49, E1L3N and 27A2 at the start of the experiment. SP49 and E1L3N which recognize the intracellular domain worked for WB. However, low level of PD-L1 was difficult to be detected with these clones by WB. 27A2 which recognizes extracellular domain did not work for WB but worked well for flow cytometry and could detect low level of PD-L1. Therefore, we investigated the PD-L1 expression by flow cytometry with 27A2 in this study. Recently, there are many commercially available PD-L1 antibodies which work for WB. In the future, we would like check the PD-L1 expression using those antibodies.